# Red Blood Cell Distribution Width: A Risk Factor for Prognosis in Patients with Ischemic Cardiomyopathy after Percutaneous Coronary Intervention

**DOI:** 10.3390/jcm12041584

**Published:** 2023-02-16

**Authors:** Biyang Zhang, Yinxiao Xu, Xin Huang, Tienan Sun, Meishi Ma, Zheng Chen, Yujie Zhou

**Affiliations:** Department of Cardiology, Capital Medical University Affiliated Anzhen Hospital, Beijing 100089, China

**Keywords:** ischemic cardiomyopathy, red blood cell distribution width, heart failure, percutaneous coronary intervention, major adverse cardiovascular events

## Abstract

Background: It has been demonstrated in previous studies that red blood cell distribution width (RDW) is correlated with the severity and prognosis of cardiovascular disease. The target of our study was to assess the relationship between RDW and the prognosis of ischemic cardiomyopathy (ICM) patients undergoing percutaneous coronary intervention (PCI). Methods: The study retrospectively enrolled 1986 ICM patients undergoing PCI. The patients were divided into three groups by RDW tertiles. The primary endpoint was major adverse cardiovascular events (MACE) and the secondary endpoints were each of the components of MACE (all-cause mortality, nonfatal myocardial infarction (MI) and any revascularization). Kaplan–Meier survival analyses were conducted to show the association between RDW and the incidence of adverse outcomes. The independent effect of RDW on adverse outcomes was determined by multivariate Cox proportional hazard regression analysis. In addition, the nonlinear relationship between RDW values and MACE was explored using restricted cubic spline (RCS) analysis. The relationship between RDW and MACE in different subgroups was determined using subgroup analysis. Results: As RDW tertiles increased, the incidences of MACE (Tertile 3 vs. Tertile 1: 42.6 vs. 23.7, *p* < 0.001), all-cause death (Tertile 3 vs. Tertile 1: 19.3 vs. 11.4, *p* < 0.001) and any revascularization (Tertile 3 vs. Tertile 1: 20.1 vs. 14.1, *p* < 0.001) increased significantly. The K–M curves showed that higher RDW tertiles were related to increased incidences of MACE (log-rank, *p* < 0.001), all-cause death (log-rank, *p* < 0.001) and any revascularization (log-rank, *p* < 0.001). After adjusting for confounding variables, RDW was proved to be independently associated with increased risks of MACE (Tertile 3 vs. Tertile 1: HR, 95% CI: 1.75, 1.43–2.15; *p* for trend < 0.001), all-cause mortality (Tertile 3 vs. Tertile 1: HR, 95% CI: 1.58, 1.17–2.13; *p* for trend < 0.001) and any revascularization (Tertile 3 vs. Tertile 1: HR, 95% CI: 2.10, 1.54–2.88; *p* for trend < 0.001). In addition, the RCS analysis suggested nonlinear association between RDW values and MACE. The subgroup analysis revealed that elderly patients or patients with angiotensin receptor blockers (ARBs) had a higher risk of MACE with higher RDW. Patients with hypercholesterolemia or without anemia also had a higher risk of MACE. Conclusions: RDW was significantly related to the increased risk of MACE among ICM patients undergoing PCI.

## 1. Introduction

With the incidence rate increasing year by year, heart failure (HF) has become a grim global public health issue [1,2]. Most patients with poor prognosis expect to live less than 5 years. Ischemic cardiomyopathy (ICM) is considered to be one of the most important causes of HF [3]. ICM refers to severe coronary artery stenosis caused by a variety of factors, resulting in a series of physiological changes. In addition, the changes may affect myocardial systolic or diastolic function, ultimately leading to HF. Although many studies on ICM having been carried out in recent years, there is still a lack of relevant research on the laboratory indicators affecting the prognosis of ICM patients [4,5].

Red blood cell distribution width (RDW), an integral component of standard complete blood count reporting, reflects the variability of red blood cell size [6]. It is well known that RDW is widely used in the clinical diagnosis and prognosis of hematological diseases [7,8]. In recent years, an increasing number of studies have indicated that there is a positive relationship between high RDW values and an increased risk of cardio-cerebrovascular diseases, such as stroke [9], non-ST-elevation myocardial infarction (MI) [10], ST-elevation MI [11,12] and stable coronary artery disease (CAD) [13]. Meanwhile, numerous studies claimed that higher RDW levels were significantly associated with morbidity and mortality [14,15] in HF patients, including patients with acute HF [16] and chronic HF [17]. Potential mechanisms to explain the correlation between RDW and ICM might be the activation of an inflammatory reaction [15,18] and impaired iron metabolism [19]. As a special cohort, ICM patients could be viewed as having CAD complicated with HF to a certain extent. Nevertheless, no relevant studies have revealed a significant association between RDW and adverse outcomes in ICM patients. As a consequence, we focused on ICM patients for the first time to explore the relationship between RDW and adverse outcomes.

## 2. Materials and Methods

### 2.1. Study Population

This study was a single-center, observational, retrospective cohort study of patients with ICM who were electively treated with PCI at Beijing Anzhen Hospital from June 2017 to June 2019. The diagnosis of patients with ICM was based on the following criteria [20]: (1) HF diagnosis based on International Classification of Diseases (ICD) 10th edition (details are listed in Appendix A); (2) concomitant multivessel disease (MVD) (coronary artery stenosis >50% in ≥2 vessels or left main). In total, we enrolled 3161 patients at our heart center. Key exclusion criteria were as follows: (1) lost to follow-up; (2) receiving coronary artery bypass grafting (CABG); (3) any cancer affecting survival; (4) left ventricular ejection fraction (LVEF) ≥ 50%; (5) RDW data missing; (6) acute myocardial infarction (MI). A total of 1986 patients were included in the final analysis (Figure 1).

### 2.2. Data Collection

All data were selected from the electronic medical recording system of Beijing Anzhen Hospital. The following data were collected: demographics, vital signs, New York Heart Association (NYHA) class, comorbidities, medical history, laboratory parameters, echocardiography, medication, angiographic data and procedural results (details are listed in Appendix A).

### 2.3. Grouping and Outcomes

All participants were classified into three groups according to tertiles of RDW (RDW arranged from small to large and divided into three equal parts): RDW < 12.9 (n = 769), 12.9 ≤ RDW < 13.5 (n = 626), RDW ≥ 13.5 (n = 591). The primary endpoint was major adverse cardiovascular events (MACE) and the secondary endpoints were each of the components of MACE (all-cause mortality, nonfatal MI and any revascularization).

### 2.4. Follow-Up

After discharge, participants were routinely followed up every 3 months for up to 12 months and then yearly for up to 36 months, by telephone questionnaire or outpatient visit. Follow-up was discontinued if death occurred. During each participant’s follow-up, the most adverse endpoint was selected for analysis (all-cause mortality > nonfatal MI > any revascularization) among multiple adverse endpoints. If a single adverse event occurred more than once during the follow-up period, only the first instance of that outcome was selected for the current study.

### 2.5. Statistical Analysis

Baseline characteristics were summarized as mean ± standard deviation (SD) for normally distributed quantitative data, as median (interquartile range (IQR)) for skewed data and as numbers (percentage) for categorical data. The ANOVA test, Mann–Whitney U test and Chi-square test were applied to compare differences between different RDW tertiles. Kaplan–Meier survival analyses were performed to assess 3-year incidences of adverse events among different groups, and the log-rank test was calculated. Cox proportional hazard regression analysis was performed to explore the association between RDW and the adverse outcomes. The results were expressed by the hazard ratio (HR) and the 95% confidence interval (CI). In Model 1, no variables were adjusted. In Model 2, age and sex were adjusted. In addition, Model 3 was adjusted for age, sex, heart rate, body mass index (BMI), NYHA class, prior percutaneous coronary intervention (PCI), platelet, total cholesterol (TC), glucose, low-density lipoprotein cholesterol (LDL-C), high-density lipoprotein cholesterol (HDL-C), potassium, angiotensin receptor blocker (ARB), sacubitril/valsartan, chronic total occlusion (CTO), diffuse lesion, left main artery (LM) disease, in-stent restenosis, SYNTAX score, target vessel of LM, diabetes, chronic kidney disease, atrial fibrillation, prior myocardial infarction (MI) and anemia. The confounding variables in Model 3 were obtained using stepwise method with removal at *p* > 0.05 and clinical doubt. In addition, based on Model 3, restricted cubic spline (RCS) analysis was conducted to explore the association between RDW as a continuous scale and the incidence of MACE.

Statistical significance was set at *p* < 0.05. Statistical analyses were performed using R software (R-project^®^; R Foundation for Statistical Computing, Vienna, Austria, ver. 4.2.1).

## 3. Results

### 3.1. Subjects and Baseline Characteristics

A total of 1986 patients were finally enrolled. The participants were divided into three groups according to the level of RDW (RDW was arranged from small to large and divided into three equal parts): RDW < 12.9 (n = 769), 12.9 ≤ RDW < 13.5 (n = 626) and 13.5 ≤ RDW (n = 591). The baseline characteristics of each group are shown in Table 1. Patients with high levels of RDW were older, more often women and more often had a history of prior stroke and/or prior MI. In addition, they had more comorbidities, such as atrial fibrillation, hypercholesterolemia, anemia and chronic kidney disease. Additionally, patients with higher tertiles of RDW had higher creatinine, blood nitrogen, BNP and RDW but lower red blood cell counts, hemoglobin, eGFR, glucose, ALT (alanine transaminase) and AST (aspartate transaminase). The left atrial diameter, LVDs and SYNTAX scores were higher, whereas LVEF was lower in patients with higher RDW. Furthermore, patients in higher RDW tertiles took more clopidogrel and CCB but less clopidogrel on its own. Of note, the management of LM as a target vessel was more frequent in patients with higher RDW.

### 3.2. Association between RDW and Adverse Outcomes

As shown in Table 2, the incidence of MACE was 33.1%. As RDW tertiles increased, the incidence of MACE increased significantly (Tertile 3 vs. Tertile 1: 42.6 vs. 23.7, *p* < 0.001). The rates of all-cause mortality, nonfatal MI and any revascularization were 15.8%, 3.2% and 14.1%, respectively. Likewise, the higher RDW tertiles were significantly related to all-cause mortality (Tertile 3 vs. Tertile 1: 19.3 vs. 11.4, *p* < 0.001) and the rate of any revascularization increased (Tertile 3 vs. Tertile 1: 19.3 vs. 11.4, *p* < 0.001). There was no statistically significant relationship between nonfatal MI and RDW (Tertile 3 vs. Tertile 1: 3.2 vs. 2.6, *p* = 0.342). The K–M curves showed that higher RDW tertiles were associated with an increased incidence of MACE (log-rank, *p* < 0.001). Moreover, according to the K–M curves, higher RDW tertiles were demonstrated to be related to increased incidences of all-cause death (log-rank, *p* < 0.001) and any revascularization (log-rank, *p* < 0.001). However, we failed to prove that higher RDW tertiles were associated with the rate of nonfatal MI increasing (log-rank, *p* = 0.230) (Figure 2).

The independent effects of RDW on the outcomes were verified using Cox regression models (Table 3). In Model 1, a higher RDW tertile was associated with a higher risk of MACE (Tertile 3 vs. Tertile 1: HR, 95% CI: 2.04, 1.69–2.47; *p* for trend < 0.001), all-cause mortality (Tertile 3 vs. Tertile 1: HR, 95% CI: 1.91, 1.45–2.52; *p* for trend < 0.001) and any revascularization (Tertile 3 vs. Tertile 1: HR, 95% CI: 2.37, 1.77–3.17; *p* for trend < 0.001). When RDW was examined as a continuous variable, each rising unit was significantly related to an increased risk of MACE. In Model 2, after adjusting for sex and age, the results were consistent with those of Model 1. In Model 3, in which more possible confounding variables were incorporated, higher RDW tertiles were still associated with increased risks of MACE (Tertile 3 vs. Tertile 1: HR, 95% CI: 1.75, 1.43–2.15; *p* for trend < 0.001), all-cause mortality (Tertile 3 vs. Tertile 1: HR, 95% CI: 1.58, 1.17–2.13; *p* for trend < 0.001) and any revascularization (Tertile 3 vs. Tertile 1: HR, 95% CI: 2.10, 1.54–2.88; *p* for trend < 0.001). When considered as a continuous variable, higher RDW was still independently correlated with increased risks of MACE (HR, 95% CI: 1.16, 1.10–1.23; *p* < 0.001), all-cause mortality (HR, 95% CI: 1.15, 1.05–1.26; *p* = 0.003) and any revascularization (HR, 95% CI: 1.19, 1.10–1.28; *p* < 0.001). However, higher RDW was not associated with an increased risk of nonfatal MI in any model (Table 3). In Figure 3, we conducted RCS analysis to analyze the nonlinear relationship between MACE and RDW as a continuous variable. After the potential confounders were taken into account, a high RDW value was positively correlated with an increased risk of MACE (nonlinear, *p* < 0.001).

### 3.3. Subgroup Analysis

No significant interactions were observed in most subgroups except in age, ARBs, anemia and hypercholesterolemia. The risk of MACE increased in elderly patients (*p* for interaction = 0.037). In addition, patients who received ARB therapy had a higher risk of MACE for RDW (*p* for interaction = 0.026) (Table 4). Patients with hypercholesterolemia (*p* for interaction = 0.011) and without anemia (*p* for interaction < 0.001) also had a higher risk of MACE.

## 4. Discussion

Our retrospective study confirmed that there was a significant association between RDW and MACE in ICM patients undergoing PCI. (1) With an increase in RDW, incidences of MACE, all-cause death and any revascularization increased significantly. (2) The K–M curves also showed that higher RDW tertiles were associated with higher incidences of MACE, all-cause death and any revascularization. (3) After adjusting for possible confounding variables, higher RDW was independently associated with increased risks of MACE, all-cause mortality and any revascularization. (4) The RCS analysis demonstrated a positive correlation between RDW and MACE. (5) Significant interactions were found in the subgroups of age and ARBs.

The correlation between RDW and ICM has a certain pathophysiological mechanism. Elevated RDW could be a marker of inflammation [15,18]. Erythrocyte production in the bone marrow is inhibited by proinflammatory cytokines, as a result of impaired erythropoietic progenitor cell proliferation and erythropoietic cell maturation [21]. In addition, reactive oxygen species generated by inflammation are associated with deranged hematopoiesis, which causes anisocytosis [22]. All of these mechanisms lead to elevations in RDW. Moreover, inflammation can increase RDW levels by altering iron metabolism. Specifically, inflammatory cytokine (IL-6) causes the overproduction of hepcidin (the key regulator of iron metabolism), which impairs the ability to mobilize and use stored iron [19]. Of note, inflammatory stress and impaired iron metabolism directly contribute to disease progression in CAD and HF [17,23]. This might be the reason why elevated RDW was associated with poor prognosis in ICM patients.

As a new indicator in recent years, RDW has been confirmed as being markedly associated with the diagnosis and prognosis of cardiovascular diseases. A retrospective cohort study involving 978 patients with acute HF showed that the postdischarge mortality rate after 31 months of follow-up was 48% and higher RDW was positively correlated with the increased risk of postdischarge mortality [24]. Another study from Yao L et al. [25] also revealed that the levels of RDW were closely related with the prognosis of patients with acute attacks of chronic HF. Another study from Nagula *p* et al. [26], which enrolled 576 patients, showed that the levels of RDW were strongly associated with the severity of CAD. In addition, a cohort study from Xiao Q et al. [27] showed that high levels of preoperative RDW increased the risk of MACE in patients who underwent PCI. Therefore, for patients with a high risk of MACE, quantitative assessment of the extent of RDW is of great clinical importance for risk stratification and prognosis prediction.

Our study also showed the similar result that higher RDW tertiles were positively correlated with the risk of MACE. In relevant studies of cardiovascular disease, this is the first time that the relationship between RDW and MACE has been illustrated through an RCS curve. Based on the analysis of the RCS curve, we found that RDW is positively correlated with MACE. The subgroup showed that elderly patents had a higher risk of MACE because of their increased risk of being in poor physical condition and more possible comorbidities with rising age, which matched our expectations. Interestingly, we found that there was an increased risk of MACE in patients who received ARB therapy and patients without anemia. It is well known that ARBs can improve the prognosis and anemia can worsen the prognosis in HF patients [28,29]. Controversy over the results of our study might arise due to the statistical bias caused by the small sample sizes of patients receiving ARB treatment (n = 223) and patients without anemia (n = 202). Therefore, further studies are needed to verify those results. Similar to several previous studies [30,31], the grouping of our study was based on the tertiles of RDW, which distribute the patients in each group more equally to avoid the statistical bias caused by small sample size.

In previous studies, ICM patients were an often-overlooked group. Compared with other cardiovascular disease patients in other studies, ICM patients can be regarded as patients with HF and coronary heart disease. Our study focused on ICM patients undergoing PCI and indicated that RDW was a risk factor for poor prognosis in ICM patients, especially elderly patients. For the sake of eliminating the effect of potential confounding variables, we conducted Cox multivariate regression analysis to verify the results. In addition, we found that as RDW increased in value, the risk of MACE increased, which may cause psychological, physical and financial burdens on patients. Therefore, clinicians should pay more attention to cheap, easily accessible predictors such as RDW. In practice, RDW provides insights for a cost-effective clinical index to judge the prognosis of ICM patients, even in some remote areas.

## 5. Limitations

(1) This was a single-center retrospective cohort study, hence there were risks of bias and lack of reproducibility that might affect the validity of the conclusions. (2) We did not find an association between RDW and nonfatal MI because of the low incidence of the latter. (3) We did not observe RDW values dynamically. (4) Almost all subjects were Chinese.

In further studies, a multicenter prospective cohort study will be conducted to filter relevant key variables in the beginning to reduce bias. We will set more specific criteria for study inclusion, such as the quality of blood and of blood flow, and blood diseases, especially anemia, thrombocytopenia, dyslipidemia and chronic hepatitis. In addition, we will exclude patients who are receiving therapeutic treatments that may affect their RDW values. Moreover, we will measure RDW values dynamically to confirm whether lowering RDW can improve prognosis. In addition, more research will be needed to determine whether this conclusion is applicable to people from other ethnic backgrounds.

## 6. Conclusions

Higher RDW was significantly related to increased risks of MACE, all-cause mortality and any revascularization among ICM patients undergoing PCI, suggesting that RDW is a risk factor for prognosis. Additionally, when examined as a continuous variable, there was a positive nonlinear relationship between RDW and MACE.

## Figures and Tables

**Figure 1 jcm-12-01584-f001:**
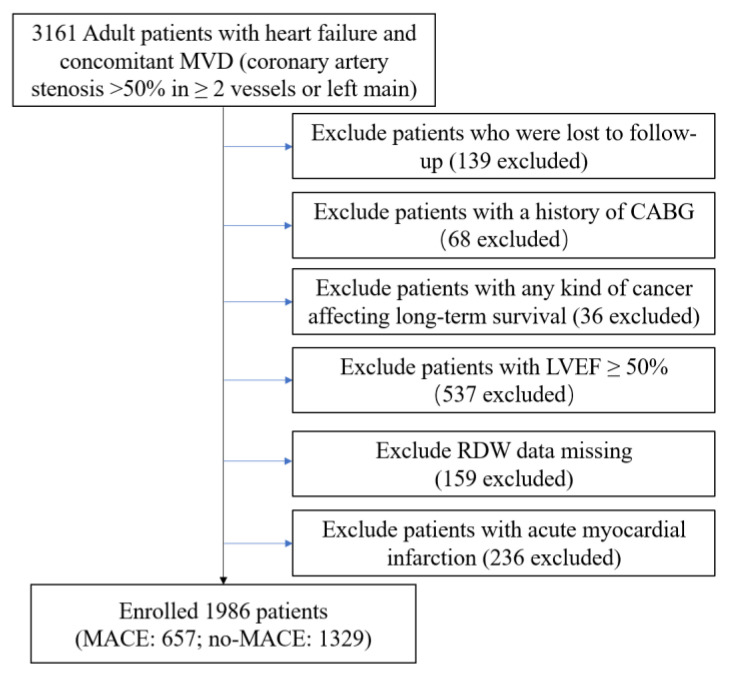
Flow chart of study population. Abbreviations: MVD: multivessel disease; CABG: coronary artery bypass grafting; LVEF: left ventricular ejection fraction; RDW: red blood cell distribution width; MACE: major adverse cardiovascular events.

**Figure 2 jcm-12-01584-f002:**
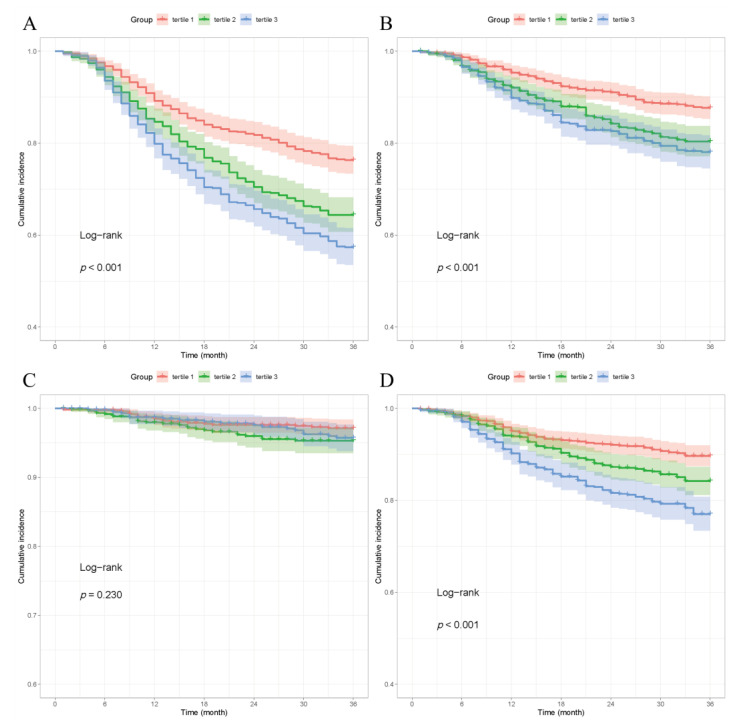
(**A**). Kaplan–Meier curves showing the association between the RDW tertiles and MACE. (**B**). Kaplan–Meier curves showing the association between the RDW tertiles and all-cause mortality. (**C**). Kaplan–Meier curves showing the association between the RDW tertiles and nonfatal MI. (**D**). Kaplan–Meier curves showing the association between the RDW tertiles and any revascularization. Abbreviations: RDW: red blood cell distribution width; MACE: major adverse cardiovascular events; MI: myocardial infarction.

**Figure 3 jcm-12-01584-f003:**
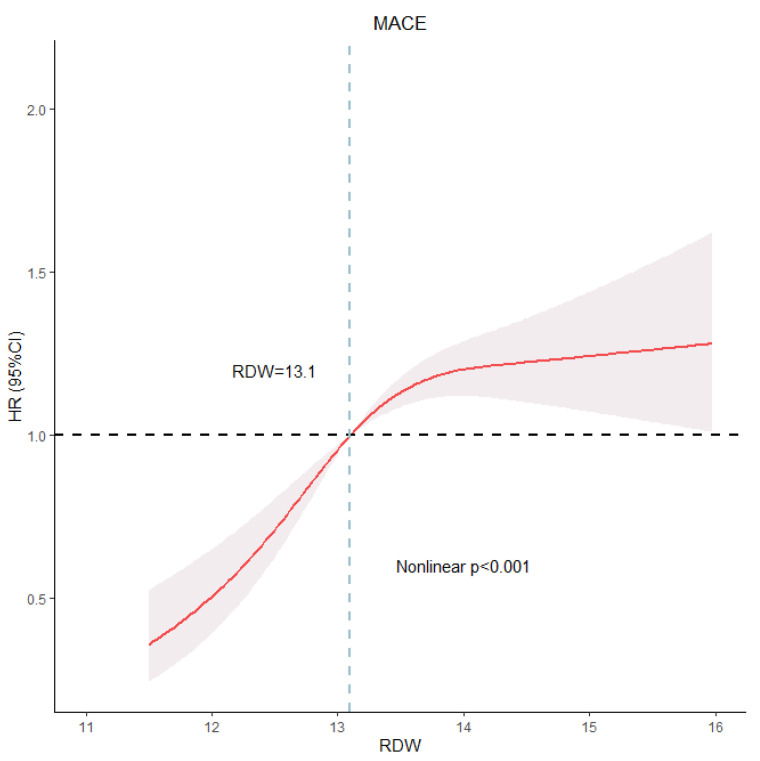
RCS model showing the association between the RDW and MACE. Abbreviations: RCS: restricted cubic spline; RDW: red blood cell distribution width; MACE: major adverse cardiovascular events; HR: hazards ratio; CI: confidence interval.

**Table 1 jcm-12-01584-t001:** Characteristics of patients stratified by RDW tertiles.

Characteristics	Total (n = 1986)	Tertiles of RDW			
		Tertile 1 (n = 769) RDW <12.9	Tertile 2 (n = 626) 12.9 ≤ RDW < 13.5	Tertile 3 (n = 591) RDW ≥ 13.5	*p* Value
**Age (years)**	60.3 ± 11.0	57.8 ± 10.5	60.5 ± 11.4	63.2 ± 10.4	<0.001
**Sex, n (%)**					0.005
Male	1690 (82.2)	666 (86.6)	5165 (82.4)	451 (76.3)	
Female	365 (17.8)	103 (13.4)	110 (17.6)	140 (23.7)	
**Vital signs**					
Systolic blood pressure (mmHg)	121.7 ± 18.2	120.6 ± 17.8	122.8 ± 17.9	122.0 ± 18.8	0.068
Diastolic blood pressure (mmHg)	73.4 ± 12.1	73.2 ± 11.9	74.0 ± 11.6	73.0 ± 12.8	0.297
Heart rate (beats/min)	73.7 ± 10.9	73.5 ± 10.3	73.5 ± 10.8	74.2 ± 11.5	0.401
Body mass index (kg/m^2^)	25.7 ± 3.4	25.7 ± 3.3	25.8 ± 3.4	25.6 ± 3.6	0.544
**NYHA class, n (%)**					0.851
I	223 (11.2)	94 (12.2)	67 (10.7)	62 (10.5)	
II	1038 (52.3)	405 (52.7)	324 (51.8)	309 (52.3)	
III	657 (33.1)	244 (31.7)	216 (34.5)	197 (33.3)	
IV	68 (3.4)	26 (3.4)	19 (3.0)	23 (2.9)	
**Comorbidities, n (%)**					
Atrial fibrillation	82 (4.1)	22 (2.7)	21 (3.4)	39 (6.6)	0.001
Hypertension	1137 (57.3)	430 (56.0)	358 (57.2)	349 (59.0)	0.511
Diabetes	772 (38.9)	316 (41.1)	225 (36.0)	231 (39.1)	0.145
Hypercholesterolemia	1445 (72.8)	595 (77.4)	456 (72.8)	394 (66.7)	<0.001
Anemia	202 (10.2)	38 (4.9)	43 (6.9)	121 (20.5)	<0.001
Thrombocytopenia	23 (1.2)	7 (0.9)	8 (1.3)	8 (1.4)	0.709
Chronic kidney disease	793 (39.9)	254 (33.0)	246 (39.3)	293 (49.6)	<0.001
**History, n (%)**					
Prior stroke	177 (8.9)	52 (6.8)	46 (7.4)	79 (13.4)	<0.001
Prior MI	483 (24.3)	158 (20.6)	159 (25.4)	166 (28.1)	0.004
Prior PCI	215 (10.8)	80 (10.4)	67 (10.7)	68 (11.5)	0.804
**Laboratory parameters**					
White blood cell (10^9^/L)	8.0 ± 2.8	7.8 ± 2.4	8.1 ± 3.0	8.0 ± 3.2	0.305
Red blood cell (10^12^/L)	4.5 ± 0.6	4.6 ± 0.5	4.6 ± 0.6	4.4 ± 0.7	<0.001
Hemoglobin (g/L)	138.6 ± 18.2	142.6 ± 16.5	140.7 ± 16.7	131.1 ± 20.0	<0.001
Platelet (10^9^/L)	222.6 ± 64.8	225.8 ± 64.3	220.7 ± 64.2	220.6 ± 66.2	0.217
Creatinine (μmol/L)	86.2 ± 58.7	78.8 ± 23.6	85.1 ± 55.6	96.9 ± 86.0	<0.001
Blood nitrogen urea (mmol/L)	6.4 ± 2.9	6.2 ± 2.3	6.3 ± 2.6	7.0 ± 3.7	<0.001
eGFR (mL/min × 1.73 m^2^)	87.8 ± 20.7	91.8 ± 17.5	88.5 ± 20.3	82.0 ± 23.5	<0.001
ALT (U/L)	25 [17, 40]	27 [19, 46]	25 [17, 38]	22 [14, 33]	<0.001
AST (U/L)	23 [18, 37]	24 [18, 44]	23 [18, 38]	22 [17, 32]	<0.001
TC (mmol/L)	4.1 ± 1.1	4.1 ± 1.0	4.1 ± 1.1	4.0 ± 1.0	0.187
TG (mmol/L)	1.7 ± 1.1	1.7 ± 1.0	1.7 ± 1.1	1.7 ± 1.1	0.390
LDL-C (mmol/L)	2.4 ± 0.9	2.5 ± 0.9	2.4 ± 0.9	2.4 ± 0.9	0.054
HDL-C (mmol/L)	1.0 ± 0.2	1.0 ± 0.2	1.0 ± 0.2	1.0 ± 0.3	0.358
Glucose	7.3 ± 3.0	7.5 ± 3.2	7.2 ± 3.0	7.0 ± 2.8	0.007
Sodium (mmol/L)	138.9 ± 3.0	138.9 ± 2.9	139.0 ± 2.8	139.0 ± 3.2	0.541
Potassium (mmol/L)	4.2 ± 0.4	4.2 ± 0.4	4.1 ± 0.4	4.2 ± 0.5	0.322
BNP (pg/mL)	337 [149, 486]	292 [119, 443]	313.5 [129, 457]	397 [207, 627]	<0.001
RDW	13.2 ± 1.1	12.4 ± 0.3	13.1 ± 0.2	14.3 ± 1.3	<0.001
**Echocardiography**					
Left atrial diameter (mm)	39.2 ± 5.2	38.4 ± 4.9	39.0 ± 4.9	40.4 ± 5.7	<0.001
LVDs (mm)	41.2 ± 8.0	40.7 ± 7.8	40.9 ± 7.9	42.1 ± 8.4	0.005
LVDd (mm)	54.7 ± 7.0	54.5 ± 6.9	54.6 ± 6.8	55.3 ± 7.5	0.102
LVEF (%)	41.0 ± 6.4	41.3 ± 6.0	41.2 ± 6.4	40.3 ± 6.8	0.001
**Medication use, n (%)**					
Aspirin	1979 (99.7)	765 (99.5)	623 (99.5)	591 (100.0)	0.224
Clopidogrel	1599 (80.5)	606 (78.8)	491 (78.4)	502 (84.9)	0.005
Ticagrelor	386 (19.4)	163 (21.2)	135 (21.6)	88 (14.9)	0.004
Statins	1972 (99.3)	762 (99.1)	622 (99.4)	588 (99.5)	0.660
CCB	241 (12.1)	75 (9.8)	73 (11.7)	93 (15.7)	0.003
Beta-blockers	1197 (60.3)	452 (55.8)	383 (61.2)	362 (61.3)	0.557
ACEI	173 (8.7)	74 (9.6)	53 (8.5)	46 (7.8)	0.475
ARB	223 (11.2)	81 (10.5)	63 (10.1)	79 (13.4)	0.140
Diuretics	1332 (67.1)	500 (65.0)	419 (66.9)	413 (69.9)	0.167
Sacubitril/valsartan	654 (32.9)	254 (33.0)	214 (34.2)	186 (31.5)	0.601
**Angiographic data, n (%)**					
LM disease	364 (18.3)	125 (16.3)	114 (18.2)	125 (21.1)	0.069
Three-vessel disease	1122 (56.5)	420 (54.6)	358 (57.2)	344 (58.2)	0.381
CTO	545 (27.4)	200 (26.0)	180 (28.8)	165 (27.9)	0.496
Diffuse lesion	388 (19.5)	142 (18.5)	134 (21.4)	112 (19.0)	0.353
In-stent restenosis	82 (4.1)	29 (3.8)	29 (4.6)	24 (4.1)	0.720
SYNTAX score	21.8 ± 7.8	21.1 ± 7.6	21.3 ± 7.8	22.4 ± 8.0	<0.001
**Procedural results, n (%)**					
Target vessel territory					
LM	330 (16.6)	116 (15.1)	103 (16.5)	111 (18.8)	0.191
LAD	1492 (75.1)	553 (71.9)	493 (78.8)	446 (75.5)	0.013
LCX	1272 (64.1)	493 (64.1)	401 (64.1)	378 (64.0)	0.998
RCA	1377 (69.3)	493 (64.1)	401 (64.1)	378 (64.0)	0.889
Complete revascularization	1216 (61.2)	465 (60.5)	397 (63.4)	354 (59.9)	0.388
Number of stents	3.3 ± 1.5	3.3 ± 1.5	3.4 ± 1.4	3.4 ± 1.5	0.599

Normally distributed variables are presented as mean ± SD. Skewed variables are presented as median (interquartile range (IQR)). Categorical variables are presented as numbers (percentage). Abbreviations: NYHA: New York Heart Association; MI: myocardial infarction; PCI: percutaneous coronary intervention; eGFR: estimated glomerular filtration rate; ALT: alanine transaminase; AST: aspartate transaminase; TC: total cholesterol; TG: triglyceride; LDL-C: low-density lipoprotein cholesterol; HDL-C: high-density lipoprotein cholesterol; BNP: brain natriuretic peptide; RDW: red blood cell distribution width; LVDs: left ventricular end systolic diameter; LVDd: left ventricular end diastolic diameter; LVEF: left ventricular injection fraction; CCB: calcium channel blocker; ACEI: angiotensin-converting enzyme inhibitor; ARB: angiotensin receptor blocker; CTO: chronic total occlusion; LM: left main artery; LAD: left anterior descending artery; LCX: left circumflex artery; RCA: right coronary artery; SYNTAX score: Syntax score is a scoring system that takes into account the location and severity of the coronary artery lesions in order to provide an overall score of the complexity of the disease.

**Table 2 jcm-12-01584-t002:** Outcomes of patients stratified by RDW tertiles.

Outcomes	Total (n = 1986)	Tertiles of RDW			
		Tertile 1 (n = 769) RDW < 12.9	Tertile 2 (n = 626) 12.9 ≤ RDW < 13.5	Tertile 3 (n = 591) RDW ≥ 13.5	*p* Value
MACE, n (%)	657 (33.1)	182 (23.7)	223 (35.6)	252 (42.6)	<0.001
All-cause mortality	313 (15.8)	88 (11.4)	111 (17.7)	114 (19.3)	<0.001
Nonfatal MI	64 (3.2)	20 (2.6)	25 (4.0)	19 (3.2)	0.342
Any revascularization	280 (14.1)	74 (9.6)	87 (13.9)	119 (20.1)	<0.001

The outcomes are presented as numbers (percentage). *p* values were calculated using Chi-square test to compare differences in outcomes between different RDW tertiles. Abbreviations: RDW: red blood cell distribution width; MACE: major adverse cardiovascular events; MI: myocardial infarction.

**Table 3 jcm-12-01584-t003:** The association between RDW and outcomes.

	Model 1			Model 2			Model 3		
	HR (95% CIs)	*p*	*p* for Trend	HR (95% CIs)	*p*	*p* for Trend	HR (95% CIs)	*p*	*p* for Trend
MACE			<0.001			<0.001			<0.001
Tertile 1: RDW < 12.9	1.0 (Ref)			1.0 (Ref)			1.0 (Ref)		
Tertile 2: 12.9 ≤ RDW < 13.5	1.62 (1.33–1.97)	<0.001		1.60 (1.31–1.94)	<0.001		1.48 (1.21–1.81)	<0.001	
Tertile 3: 13.5 ≤ RDW	2.04 (1.69–2.47)	<0.001		1.98 (1.63–2.41)	<0.001		1.75 (1.43–2.15)	<0.001	
Continuous	1.14 (1.09–1.18)	<0.001		1.13 (1.08–1.18)	<0.001		1.16 (1.10–1.23)	<0.001	
All-cause mortality			<0.001			<0.001			<0.001
Tertile 1: RDW < 12.9	1.0 (Ref)			1.0 (Ref)			1.0 (Ref)		
Tertile 2: 12.9 ≤ RDW < 13.5	1.67 (1.26–2.21)	<0.001		1.65 (1.24–2.18)	0.001		1.55 (1.16–2.06)	0.003	
Tertile 3: RDW ≥ 13.5	1.91 (1.45–2.52)	<0.001		1.86 (1.40–2.47)	<0.001		1.58 (1.17–2.13)	0.003	
Continuous	1.13 (1.06–1.20)	<0.001		1.12 (1.05–1.20)	0.001		1.15 (1.05–1.26)	0.003	
Nonfatal MI			0.231			0.123			0.001
Tertile 1: RDW < 12.9	1.0 (Ref)			1.0 (Ref)			1.0 (Ref)		
Tertile 2: 12.9 ≤ RDW < 13.5	1.65 (0.92–2.98)	0.094		1.63 (0.90–2.95)	0.105		1.48 (0.80–2.72)	0.211	
Tertile 3: RDW ≥ 13.5	1.40 (0.75–2.62)	0.296		1.37 (0.72–2.60)	0.331		1.29 (0.66–2.54)	0.462	
Continuous	1.06 (0.88–1.28)	0.554		1.06 (0.88–1.28)	0.560		1.07 (0.86–1.34)	0.538	
Any revascularization			<0.001			<0.001			<0.001
Tertile 1: RDW < 12.9	1.0 (Ref)			1.0 (Ref)			1.0 (Ref)		
Tertile 2: 12.9 ≤ RDW < 13.5	1.56 (1.14–2.12)	0.005		1.53 (1.12–2.08)	0.008		1.40 (1.02–1.93)	0.036	
Tertile 3: RDW ≥ 13.5	2.37 (1.77–3.17)	<0.001		2.29 (1.70–3.08)	<0.001		2.10 (1.54–2.88)	<0.001	
Continuous	1.16 (1.09–1.23)	<0.001		1.16 (1.09–1.23)	<0.001		1.19 (1.10–1.28)	<0.001	

Models were derived from Cox proportional hazards regression analysis. Model 1: unadjusted. Model 2: adjusted for age, sex. Model 3: adjusted for age, sex, heart rate, body mass index, NYHA class, prior PCI, platelet, TC, LDL-C, HDL-C, potassium, ARB, glucose, sacubitril/valsartan, diffuse lesion, SYNTAX score, LM disease, CTO, in-stent restenosis, target vessel of LM, diabetes, chronic kidney disease, atrial fibrillation, prior MI, anemia. Abbreviations: NYHA: New York Heart Association; PCI: percutaneous coronary intervention; TC: total cholesterol; LDL-C: low-density lipoprotein cholesterol; HDL-C: high-density lipoprotein cholesterol; ARB: angiotensin receptor blocker; LM: left main artery; CTO: chronic total occlusion; MI: myocardial infarction; RDW: red blood cell distribution width; MACE: major adverse cardiovascular events; HR: hazards ratio; CI: confidence interval.

**Table 4 jcm-12-01584-t004:** Subgroup analysis of associations between RDW and MACE.

Subgroups	N	HR (95% CIs)	*p*	*p* for Interaction
Age (years)				0.037
<61	941	1.12 (1.04–1.22)	0.005	
≥61	1045	1.14 (1.08–1.20)	<0.001	
Sex, n (%)				0.278
Male	1633	1.13 (1.08–1.18)	<0.001	
Female	353	1.24 (1.06–1.45)	0.008	
LM disease				0.329
Yes	364	1.21 (1.08–1.36)	0.002	
No	1622	1.13 (1.07–1.18)	<0.001	
Body mass index (kg/m^2^)				0.277
<25.3	1062	1.10 (1.04–1.18)	0.002	
≥25.3	924	1.31 (1.20–1.42)	<0.001	
Heart rate (beats/min)				0.425
<71	931	1.10 (1.04–1.17)	0.002	
≥71	1055	1.25 (1.15–1.35)	<0.001	
ARB				0.026
Yes	223	1.32 (1.16–1.52)	<0.001	
No	1763	1.12 (1.07–1.18)	<0.001	
Prior PCI				0.303
Yes	215	1.07 (0.96–1.21)	0.223	
No	1771	1.15 (1.10–1.20)	<0.001	
Glucose (mmol/L)				0.160
<6.3	966	1.23 (1.13–1.33)	<0.001	
≥6.3	1020	1.11 (1.05–1.17)	<0.001	
In-stent restenosis				0.104
Yes	82	1.81 (1.04–3.14)	0.036	
No	1904	1.13 (1.09–1.18)	<0.001	
HDL-C (mmol/L)				0.463
<1.0	1127	1.13 (1.07–1.18)	<0.001	
≥1.0	1052	1.15 (1.05–1.25)	0.002	
Sacubitril/valsartan				0.309
Yes	654	1.12 (1.04–1.20)	0.002	
No	1332	1.17 (1.10–1.24)	<0.001	
NYHA class, n (%)				0.093
I	223	1.08 (0.97–1.20)	0.159	
II	1038	1.15 (1.07–1.24)	<0.001	
III	657	1.24 (1.13–1.35)	<0.001	
IV	68	1.07 (0.77–1.50)	0.676	
SYNTAX score				0.604
<21	933	1.16 (1.07–1.25)	<0.001	
≥21	1053	1.12 (1.07–1.18)	<0.001	
TC (mmol/L)				0.507
<3.92	938	1.15 (1.07–1.24)	<0.001	
≥3.92	1048	1.13 (1.07–1.19)	<0.001	
LDL-C (mmol/L)				0.492
<2.3	1044	1.10 (1.04–1.17)	0.001	
≥2.3	942	1.26 (1.17–1.37)	<0.001	
Diffuse lesion				0.154
Yes	388	1.26 (1.09–1.46)	0.002	
No	1598	1.13 (1.08–1.19)	<0.001	
CTO				0.819
Yes	545	1.14 (1.07–1.23)	<0.001	
No	1441	1.13 (1.07–1.19)		
Potassium (mmol/L)				0.899
<4.13	930	1.13 (1.08–1.19)	<0.001	
≥4.13	1056	1.15 (1.06–1.23)	<0.001	
Target vessel of LM				0.486
Yes	330	1.20 (1.05–1.36)	0.007	
No	1656	1.13 (1.08–1.19)	<0.001	
Hypercholesterolemia				0.011
Yes	1445	1.24 (1.15–1.33)	<0.001	
No	541	1.09 (1.01–1.17)	0.022	
Anemia				<0.001
Yes	202	0.99 (0.89–1.10)	0.878	
No	1784	1.32 (1.23–1.41)	<0.001	
Thrombocytopenia				0.647
Yes	23	0.98 (0.52–1.86)	0.961	
No	1963	1.14 (1.09–1.19)	<0.001	
ALT (U/L)				0.144
<25	902	1.10 (1.04–1.17)	0.002	
≥25	1084	1.27 (1.17–1.39)	<0.001	
AST (U/L)				0.218
<23	938	1.10 (1.04–1.17)	0.001	
≥23	1048	1.25 (1.15–1.36)	<0.001	

Cox proportional hazards regression analysis was used and results are presented as HR and 95% CI. *p* for interaction was calculated using Cox proportional hazards analysis to determine whether there was interaction between different subgroups and RDW tertiles. Abbreviations: LM: left main artery; ARB: angiotensin receptor blocker; PCI: percutaneous coronary intervention; HDL-C: high-density lipoprotein cholesterol; NYHA: New York Heart Association; TC: total cholesterol; LDL-C: low-density lipoprotein cholesterol; CTO: chronic total occlusion; ALT: alanine transaminase; AST: aspartate transaminase; RDW: red blood cell distribution width; MACE: major adverse cardiovascular events; HR: hazards ratio; CI: confidence interval; SYNTAX score: Syntax score is a scoring system that takes into account the location and severity of the coronary artery lesions in order to provide an overall score of the complexity of the disease.

## Data Availability

The datasets used and/or analyzed during the current study are available from the corresponding author upon reasonable request.

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
