# Peer review of "Red Blood Cell Distribution Width: A Risk Factor for Prognosis in Patients with Ischemic Cardiomyopathy after Percutaneous Coronary Intervention"

_jcm, 2023, doi:10.3390/jcm12041584_

Round 1

Reviewer 1 Report

I have read with interest the paper produced by Biyang Zhang and colleagues and I congratulate them for the choice of the topic and for the excellent work.

Some question:

- in model 3 confounders were assessed. Why were the main risk factors for cardiovascular events not included (diabetes mellitus, CKD, AF, prior MI etc)?

- moreover, considering that patients in the third tertile took less P2Y12i, could this have any relevance? 

- Could anemia be a confounding factor?

Correct an error in line 85: "13.5≤RDW (n=591)" --> Use "≥"

Finally, as you suggested in "Limitations", relying on a single RDW value for evaluation can be confusing. It would be interesting to evaluate its value dynamically

Author Response

Dear Reviewer:

Thank you for your letter and for the reviewers’ comments concerning our manuscript entitled “Red Blood Cell Distribution Width: A risk Factor for Prognosis in Patients with Ischemic Cardiomyopathy after Percutaneous Coronary Intervention”. Those comments are all valuable and very helpful for revising and improving our paper, as well as the important guiding significance to our researches. We have studied comments carefully and have made corrections which we hope meet with approval. We uploaded the manuscript and the modified text was marked in red. The main corrections in the paper and the responds to the reviewer’s comments are as following:

Responds to the reviewer 1’s comments:

  1. Response to comment ( in model 3 confounders were assessed. Why were the main risk factors for cardiovascular events not included (diabetes mellitus, CKD, AF, prior MI etc)?)

Your comment is valuable. The confounding variables in model 3 were obtained by stepwise method with removal at P > 0.05 initially. That’s the reason why we did not include diabetes mellitus, CKD, AF and prior MI. After rethinking, we found that our original model was one-sided. We are sorry that we didn't take it into full consideration.

According to your advice, we added diabetes mellitus, AF and prior MI into model 3 and re-performed cox proportional hazard regression to explore the association between RDW and the adverse outcomes.

  1. Response to comment (moreover, considering that patients in the third tertile took less P2Y12i, could this have any relevance? )

Thanks for your comments. Our study was a single-center retrospective cohort study and the sample size was not large enough. We think the relevance may be due to the statistical bias that might affect the validity of the conclusions. There was less relevance probably between patients in the third tertile and less P2Y12 therapy.

In further study, we will build a multi-center large sample database to reduce the statistical bias and abnormal results.

  1. Response to comment (Could anemia be a confounding factor?)

Thanks for your comments. We are sorry for ignoring this important clinical indicator.

  According to your advice, we added anemia into the baseline and the model 3. Cox regression was re-performed to explore the association between RDW and the adverse outcomes.

  1. Response to comment (Correct an error in line 85: "13.5≤RDW (n=591)" --> Use "≥")

Thanks for your careful review. We are very sorry for the careless clerical error. We corrected the error in the article.

Special thanks to you for your good comments.

We tried our best to improve the manuscript and made some changes in the manuscript. These changes will not influence the content and framework. The manuscript was uploaded and the modified text was marked in red.

We appreciate for Editors and Reviewers’ warm work earnestly, and hope that the correction will meet with approval.

Once again, thank you very much for your comments and suggestions. And if you have any queries, please don't hesitate to contact me.

Thank you and best regards.

We would like to thank the editors and reviewers for their enthusiastic work and look forward to your reply

Thank you again for your work. If you have any questions, please do not hesitate to contact me.

The revised version of manuscript has upload in an attatchment.

Reviewer 2 Report

Rows 21-24 to be reformulated, with no tertile’s  specific description, this will be interpreted and presented in results and discussions sections.

“The relationship between RDW and MACE in different subgroups was determined by sub- 21

group analysis. Result: As RDW tertiles increased, the incidence of MACE (Tertile 3 vs. Tertile 1: 22

42.6 vs. 23.7, P <0.001), all-cause death (Tertile 3 vs. Tertile 1: 19.3 vs. 11.4, P< 0.001) and any revascularization (Tertile 3 vs. Tertile 1: 20.1 vs. 14.1, P< 0.001) increased significantly.”

Same comment for rows 27-31

“. After adjusting for confounding variables, RDW was proved to be independently associated 27

with the increased risk of MACE (Tertile 3 vs Tertile 1: HR, 95% CI: 1.79, 1.47-2.19; P for trend 28

<0.001), all-cause mortality (Tertile 3 vs Tertile 1: HR, 95% CI: 1.66, 1.24-2.23; P for trend <0.001) and 29

any revascularization (Tertile 3 vs Tertile 1: HR, 95% CI: 2.09,1.55-2.84; P for trend <0.001) respec- 30

tively.”

Introduction must be improved, with literature references and consistent argumentation. Please do not attribute so many reference titles all together,  because it rises the supposition it is not a very accurate work.

Informations about other blood modificationsm such as blood flow, dyslipidemia, drugs that affect blood caracteristics are missing. Maybe rows like 223-234 colud be useful in introduction, also.

Example: .” Rows 50-52, needs to be wiser explained and adequately referred.

Affirmations like : ” In recent years, the increasing studies have indicated that there was a positive relationship between high values of RDW and increased risk of adverse events in patients with cardio-cerebrovascular disease

Material and method:  one of the criteria should be the quality of the blood and of the blood flow. What about blood diseases, such as anemia, thrombocytopenia , dyslipidemia, chronic hepatitis in the exclusion criteria?

The Follow-up, rows 88-93, should be reformulated and much worse, rethink.

Author Response

Dear Reviewer:

Thank you for your letter and for the reviewers’ comments concerning our manuscript entitled “Red Blood Cell Distribution Width: A risk Factor for Prognosis in Patients with Ischemic Cardiomyopathy after Percutaneous Coronary Intervention”. Those comments are all valuable and very helpful for revising and improving our paper, as well as the important guiding significance to our researches. We have studied comments carefully and have made corrections which we hope meet with approval. We uploaded the manuscript and the modified text was marked in red. The main corrections in the paper and the responds to the reviewer’s comments are as following:

Responds to the reviewer 2’s comments:

1.Response to comment (Rows 21-24 to be reformulated, with no tertiles  specific description, this will be interpreted and presented in results and discussions sections.The relationship between RDW and MACE in different subgroups was determined by subgroup analysis. Result: As RDW tertiles increased, the incidence of MACE (Tertile 3 vs. Tertile 1: 42.6 vs. 23.7, P <0.001), all-cause death (Tertile 3 vs. Tertile 1: 19.3 vs. 11.4, P< 0.001) and any revascularization (Tertile 3 vs. Tertile 1: 20.1 vs. 14.1, P< 0.001) increased significantly.Same comment for rows 27-31. After adjusting for confounding variables, RDW was proved to be independently associated with the increased risk of MACE (Tertile 3 vs Tertile 1: HR, 95% CI: 1.79, 1.47-2.19; P for trend <0.001), all-cause mortality (Tertile 3 vs Tertile 1: HR, 95% CI: 1.66, 1.24-2.23; P for trend <0.001) and any revascularization (Tertile 3 vs Tertile 1: HR, 95% CI: 2.09,1.55-2.84; P for trend <0.001) respectively.)

  Your advice is very valuable. Due to our negligence, tertile is not clearly defined in the article. We are very sorry for the deviation in understanding. According to your advice, we have reviewed some relevant literature and defined tertile specifically in results and discussions sections.

2.Response to comment (Introduction must be improved, with literature references and consistent argumentation. Please do not attribute so many reference titles all together, because it rises the supposition it is not a very accurate work.)

  Thanks for your comments. We are sorry to attribute so many reference titles all together, which may cause some doubts to readers. According to your advice, we have revised the introduction and attributed relevant references titles to corresponding parts more accurately.

  1. Response to comment (Information about other blood modificationsn such as blood flow, dyslipidemia, drugs that affect blood characteristics are missing. Maybe rows like 223-234 could be useful in introduction, also.)

  Thanks for your comments. The information of the hypercholesterolemia had added into the baseline. Due to the limitation of the database, we failed to get the information of blood flow and drugs which affect blood characteristics. Your advice benefits me a lot. According to your suggestion, we modified and added the relevant information of row 223-234 into the introduction.

  In further study, we will build a complete database which include more relevant information. Thank you again for your valuable advice.

  1. Response to comment Example: . Rows 50-52, needs to be wiser explained and adequately referred.Affirmations like : In recent years, the increasing studies have indicated that there was a positive relationship between high values of RDW and increased risk of adverse events in patients with cardio-cerebrovascular disease)

  Your advice is very valuable. According to your advice, we optimized the expression of relevant content and changed the reference to a more appropriate position.

  1. Response to comment (Material and method: one of the criteria should be the quality of the blood and of the blood flow. What about blood diseases, such as anemia, thrombocytopenia, dyslipidemia, chronic hepatitis in the exclusion criteria?)

  Thanks for your valuable comments. Our study mainly focused on patients with ICM. At the same time, most of these patients have blood diseases, such as anemia, thrombocytopenia, dyslipidemia and chronic hepatitis. If we exclude these people, it will lead to a significant decline in sample size, resulting in statistical bias. According to your advice, we added these factors into the baseline and subgroup analysis to explore the relationship between RDW and ICM in different diseases.

While, as the retrospective cohort study, the data of chronic hepatitis was not available. We used ALT, AST to replace hepatitis and included ALT, AST in subgroup to explore the relationship between liver function and ICM.

  1. Response to comment (The Follow-up, rows 88-93, should be reformulated and much worse, rethink.)

  Your advice is very valuable. According to your advice, we reviewed some relevant literature and tried our best to rewrite the follow-up.

Special thanks to you for your good comments.

We tried our best to improve the manuscript and made some changes in the manuscript. These changes will not influence the content and framework. The manuscript was uploaded, and the modified text was marked in red.

We appreciate for Editors and Reviewers’ warm work earnestly, and hope that the correction will meet with approval.

Once again, thank you very much for your comments and suggestions. And if you have any queries, please don't hesitate to contact me.

Thank you and best regards.

We would like to thank the editors and reviewers for their enthusiastic work and look forward to your reply.

Thank you again for your work. If you have any questions, please do not hesitate to contact me.

The revised version of manuscript has uploaded in the attachment

Round 2

Reviewer 2 Report

The authors tried to answer every inquiery, I appreciate the gesture and the content.

Further internal medicine considerations culd have been providential to the novelty and scientific soundness of the study.

I consider that having so many items, so many variables could leed to a superficial understanding of the main message of this study. I suggest, for future works, a more concise plan, which specifies from the very beginning what are the purposes and the plan of the research, the hypothesis and in the discussions, to present whether they were true or false.

Author Response

Responds to the reviewer 2’s comments:

Response to comment (I consider that having so many items, so many variables could lead to a superficial understanding of the main message of this study. I suggest, for future works, a more concise plan, which specifies from the very beginning what are the purposes and the plan of the research, the hypothesis and in the discussions, to present whether they were true or false)

Your advice is very valuable. Indeed, our study was a single-center retrospective cohort study which including many variables in final analysis, leading to a superficial understanding in results. In further study, multicenter prospective cohort study is conducted to filter relevant key variables in the beginning to reduce the bias. We will set more specific criteria for study inclusions, such as the quality of the blood and of the blood flow, blood diseases especially anemia, thrombocytopenia, dyslipidemia, chronic hepatitis. And we will exclude the patients which receive the therapeutic treatments which may affect the RDW values. Moreover, we will measure RDW values dynamically to confirm whether lowing RDW can improve the prognosis.  

According to your advice, we revised the limitation. We hope the modification can meet your standard. Thank you again for your valuable advice.

 If you have any questions, please do not hesitate to contact me.
